# Deep ReLU Networks Have Surprisingly Few Activation Patterns

**Boris Hanin**
Facebook AI Research
Texas A&M University
bhanin@math.tamu.edu

**David Rolnick**
University of Pennsylvania
Philadelphia, PA USA
drolnick@seas.upenn.edu

## Abstract

The success of deep networks has been attributed in part to their expressivity: per parameter, deep networks can approximate a richer class of functions than shallow networks. In ReLU networks, the number of activation patterns is one measure of expressivity; and the maximum number of patterns grows exponentially with the depth. However, recent work has showed that the practical expressivity of deep networks – the functions they can learn rather than express – is often far from the theoretical maximum. In this paper, we show that the average number of activation patterns for ReLU networks at initialization is bounded by the total number of neurons raised to the input dimension. We show empirically that this bound, which is independent of the depth, is tight both at initialization and during training, even on memorization tasks that should maximize the number of activation patterns. Our work suggests that realizing the full expressivity of deep networks may not be possible in practice, at least with current methods.

## 1  Introduction

A fundamental question in the theory of deep learning is why deeper networks often work better in practice than shallow ones. One proposed explanation is that, while even shallow neural networks are universal approximators [3, 7, 9, 16, 21], there are functions for which increased depth allows exponentially more efficient representations. This phenomenon has been quantified for various complexity measures [4, 5, 6, 10, 18, 19, 22, 23, 24, 25]. However, authors such as Ba and Caruana have called into question this point of view [2], observing that shallow networks can often be trained to imitate deep networks and thus that functions learned in practice by deep networks may not achieve the full expressive power of depth.

In this article, we attempt to capture the difference between the maximum complexity of deep networks and the complexity of functions that are actually learned (see Figure 1). We provide theoretical and empirical analyses of the typical complexity of the function computed by a ReLU network $\mathcal{N}$. Given a vector $\theta$ of its trainable parameters, $\mathcal{N}$ computes a continuous and piecewise linear function $x \mapsto \mathcal{N}(x; \theta)$. Each $\theta$ thus is associated with a partition of input space $\mathbb{R}^{n_{\text{in}}}$ into activation regions, polytopes on which $\mathcal{N}(x; \theta)$ computes a single linear function corresponding to a fixed activation pattern in the neurons of $\mathcal{N}$.

We aim to count the number of such activation regions. This number has been the subject of previous work (see §1.1), with the majority concerning large *lower bounds* on the *maximum* over all $\theta$ of the number of regions for a given network architecture. In contrast, we are interested in the typical behavior of ReLU nets as they are used in practice. We therefore focus on small *upper bounds* for the *average* number of activation regions present for a typical value of $\theta$. Our main contributions are:

- We give precise definitions and prove several fundamental properties of both linear and activation regions, two concepts that are often conflated in the literature (see §2).
- We prove in Theorem 5 an upper bound for the expected number of activation regions in a ReLU net $\mathcal{N}$. Roughly, we show that if $n_{\text{in}}$ is the input dimension and $\mathcal{C}$ is a cube in input

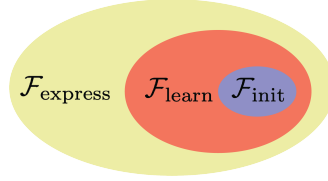

Figure 1: Schematic illustration of the space of functions $f : \mathbb{R}^{n_{in}} \to \mathbb{R}^{n_{out}}$. For a given neural network architecture, there is a set $\mathcal{F}_{express}$ of functions expressible by that architecture. Within this set, the functions corresponding to networks at initialization are concentrated within a set $\mathcal{F}_{init}$. Intermediate between $\mathcal{F}_{init}$ and $\mathcal{F}_{express}$ is a set $\mathcal{F}_{learn}$ containing the functions which the network has non-vanishing probability of learning using gradient descent. (None of these is of course a formal definition.) This paper seeks to demonstrate the gap between $\mathcal{F}_{express}$ and $\mathcal{F}_{learn}$ and that, at least for certain measures of complexity, there is a surprisingly small gap between $\mathcal{F}_{init}$ and $\mathcal{F}_{learn}$.

space $\mathbb{R}^{n_{in}}$, then, under reasonable assumptions on network gradients and biases,

$$\frac{\#\text{activation regions of } \mathcal{N} \text{ that intersect } \mathcal{C}}{\text{vol}(\mathcal{C})} \leq \frac{(T\#\text{neurons})^{n_{in}}}{n_{in}!}, \quad T > 0. \quad (1)$$

- This bound holds in particular for deep ReLU nets at initialization, and is in sharp contrast to the maximum possible number of activation patterns, which is exponential in depth [23, 28].

- Theorem 5 also strongly suggests that the bounds on number of activation regions continue to hold approximately throughout training. We empirically verify that this behavior holds, even for networks trained on memorization-based tasks (see §4 and Figures 3-6).

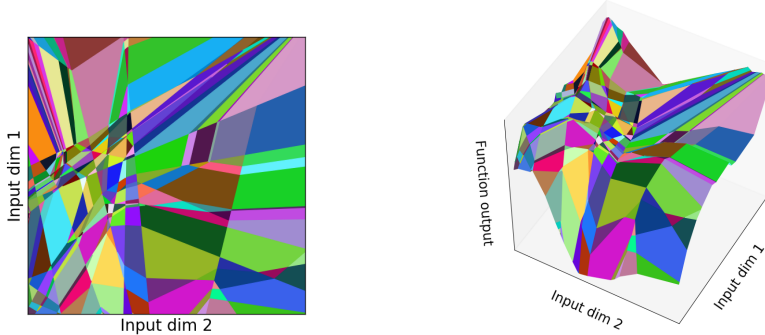

Figure 2: Function defined by a ReLU network of depth 5 and width 8 at initialization. Left: Partition of the input space into regions, on each of which the activation pattern of neurons is constant. Right: the function computed by the network, which is linear on each activation region.

It may seem counterintuitive that the number of activation patterns in a ReLU net is effectively capped far below its theoretical maximum during training, even for tasks where a higher number of regions would be advantageous (see §4). We provide in §3.2-3.3 two intuitive explanations for this phenomenon. The essence of both is that many activation patterns can be created only when a typical neuron $z$ in $\mathcal{N}$ turns on/off repeatedly, forcing the value of its pre-activation $z(x)$ to cross the level of its bias $b_z$ many times. This requires (i) significant overlap between the range of $z(x)$ on the different activation regions of $x \mapsto z(x)$ and (ii) the bias $b_z$ to be picked within this overlap. Intuitively, (i) and (ii) require either large or highly coordinated gradients. In the former case, $z(x)$ oscillates over a large range of outputs and $b_z$ can be random, while in the latter $z(x)$ may oscillate only over a small range of outputs and $b_z$ is carefully chosen. Neither is likely to happen with a proper initialization. Moreover, both appear to be difficult to learn with gradient-based optimization.

The rest of this article is structured as follows. Section 2 gives formal definitions and some important properties of both activation regions and the closely related notion of linear regions (see Definitions 1 and 2).Section 3 contains our main technical result, Theorem 5, stated in §3.1. Sections 3.2 and 3.3 provide heuristics for understanding Theorem 5 and its implications. Finally, §4 is devoted to experiments that push the limits of how many activation regions a ReLU network can learn in practice.

## 1.1 Relation to Prior Work

We consider the typical number of activation regions in ReLU nets. Interesting bounds on the maximum number of regions are given in [4, 6, 19, 22, 23, 25, 26, 28]. Our main theoretical result,

Theorem 5, is related to [14], which conjectured that our Theorem 5 should hold and proved bounds for other notions of average complexity of activation regions. Theorem 5 is also related in spirit to [8], which uses a mean field analysis of wide ReLU nets to show that they are biased towards simple functions. Our empirical work (e.g. §4) is related both to the experiments of [20] and to those of [1, 29]. The last two observe that neural networks are capable of fitting noisy or completely random data. Theorem 5 and experiments in §4 give a counterpoint, suggesting limitations on the complexity of random functions that ReLU nets can fit in practice (see Figures 4-6).

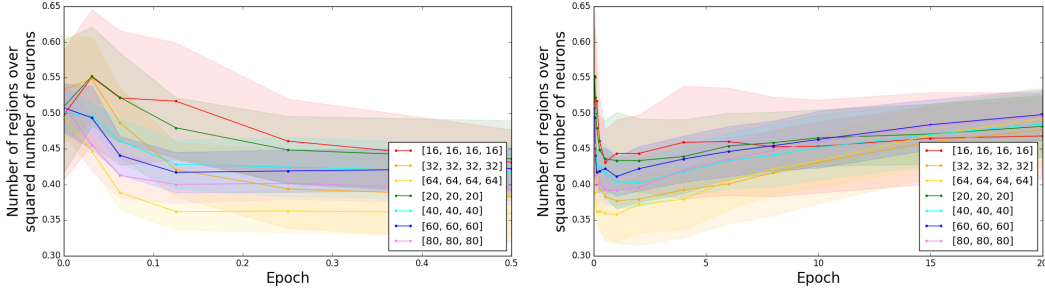

Figure 3: The average number of activation regions in a 2D cross-section of input space, for fully connected networks of various architectures training on MNIST. Left: a closeup of 0.5 epochs of training. Right: 20 epochs of training. The notation $[20, 20, 20]$ indicates a network with three layers, each of width 20. The number of activation regions starts at approximately $(\#\mathrm{neurons})^2/2$, as predicted by Theorem 5 (see Remark 1). This value changes little during training, first decreasing slightly and then rebounding, but never increasing exponentially. Each curve is averaged over 10 independent training runs, and for each run the number of regions is averaged over 5 different 2D cross-sections, where for each cross-section we count the number of regions in the (infinite) plane passing through the origin and two random training examples. Standard deviations between different runs are shown for each curve. See Appendix A for more details.

## 2 How to Think about Activation Regions

Before stating our main results on counting activation regions in §3, we provide a formal definition and contrast them with linear regions in §2.1. We also note in §2.1 some simple properties of activation regions that are useful both for understanding how they are built up layer by layer in a deep ReLU net and for visualizing them. Then, in §2.2, we explain the relationship between activation regions and arrangements of bent hyperplanes (see Lemma 4).

### 2.1 Activation Regions vs. Linear Regions

Our main objects of study in this article are activation regions, which we now define.

**Definition 1** (Activation Patterns/Regions). *Let $\mathcal{N}$ be a ReLU net with input dimension $n_{\mathrm{in}}$. An activation pattern for $\mathcal{N}$ is an assignment to each neuron of a sign:*

$$\mathcal{A} := \{a_z, \ z \text{ a neuron in } \mathcal{N}\} \in \{-1, 1\}^{\#\mathrm{neurons}}.$$

*Fix $\theta$, a vector of trainable parameters in $\mathcal{N}$, and an activation pattern $\mathcal{A}$. The activation region corresponding to $\mathcal{A}, \theta$ is*

$$\mathcal{R}(\mathcal{A}; \theta) := \{x \in \mathbb{R}^{n_{\mathrm{in}}} \mid (-1)^{a_z} (z(x; \theta) - b_z) > 0, \quad z \text{ a neuron in } \mathcal{N}\},$$

*where neuron $z$ has pre-activation $z(x; \theta)$, bias $b_z$, and post-activation $\max\{0, z(x; \theta) - b_z\}$. We say the activation regions of $\mathcal{N}$ at $\theta$ are the non-empty activation regions $\mathcal{R}(\mathcal{A}, \theta)$.*

Perhaps the most fundamental property of activation regions is their convexity.

**Lemma 1** (Convexity of Activation Regions). *Let $\mathcal{N}$ be a ReLU net. Then for every activation pattern $\mathcal{A}$ and any vector $\theta$ of trainable parameters for $\mathcal{N}$ each activation region $\mathcal{R}(\mathcal{A}; \theta)$ is convex.*

We note that Lemma 1 has been observed before (e.g. Theorem 2 in [23]), but in much of the literature the difference between linear regions (defined below), which are not necessarily convex, and activation regions, which are, is ignored. It turns out that Lemma 1 holds for *any* piecewise linear activation, such as leaky ReLU and hard hyperbolic tangent/sigmoid. This fact seems to be less

well-known (see Appendix B.1 for a proof). To provide a useful alternative description of activation regions for a ReLU net $\mathcal{N}$, a fixed vector $\theta$ of trainable parameters and neuron $z$ of $\mathcal{N}$, define

$$H_z(\theta) := \{x \in \mathbb{R}^{n_{\text{in}}} \mid z(x; \theta) = b_z\}. \tag{2}$$

The sets $H_z(\theta)$ can be thought of as "bent hyperplanes" (see Lemma 4). The non-empty activation regions of $\mathcal{N}$ at $\theta$ are the connected components of $\mathbb{R}^{n_{\text{in}}}$ with all the bent hyperplanes $H_z(\theta)$ removed:

**Lemma 2** (Activation Regions as Connected Components). *For any ReLU net $\mathcal{N}$ and any vector $\theta$ of trainable parameters*

$$\text{activation regions}\,(\mathcal{N}, \theta) \;=\; \text{connected components}\big(\mathbb{R}^{n_{\text{in}}} \setminus \bigcup_{\text{neurons } z} H_z(\theta)\big). \tag{3}$$

We prove Lemma 2 in Appendix B.2. We may compare activation regions with *linear regions*, which are the regions of input space on which the network defines different linear functions.

**Definition 2** (Linear Regions). *Let $\mathcal{N}$ be a ReLU net with input dimension $n_{\text{in}}$, and fix $\theta$, a vector of trainable parameters for $\mathcal{N}$. Define*

$$\mathcal{B}_\mathcal{N}(\theta) := \{x \in \mathbb{R}^{n_{\text{in}}} \mid \nabla \mathcal{N}\,(\cdot\,; \theta) \text{ is discontinuous at } x\}.$$

*The linear regions of $\mathcal{N}$ at $\theta$ are the connected components of input space with $\mathcal{B}_\mathcal{N}$ removed:*

$$\text{linear regions}\,(\mathcal{N}, \theta) \;=\; \text{connected components}\,(\mathbb{R}^{n_{\text{in}}} \backslash \mathcal{B}_\mathcal{N}(\theta))\,.$$

Linear regions have often been conflated with activation regions, but in some cases they are different. This can, for example, happen when an entire layer of the network is zeroed out by ReLUs, leading many distinct activation regions to coalesce into a single linear region. However, the number of activation regions is always at least as large as the number of linear regions.

**Lemma 3** (More Activation Regions than Linear Regions). *Let $\mathcal{N}$ be a ReLU net. For any parameter vector $\theta$ for $\mathcal{N}$, the number of linear regions in $\mathcal{N}$ at $\theta$ is always bounded above by the number of activation regions in $\mathcal{N}$ at $\theta$. In fact, the closure of every linear region is the closure of the union of some number of activation regions.*

Lemma 3 is proved in Appendix B.3. We prove moreover in Appendix B.4 that generically, the gradient of $\nabla \mathcal{N}$ is different in the interior of most activation regions and hence that most activation regions lie in different linear regions. In particular, this means that the number of linear regions is generically very similar to the number of activation regions.

## 2.2 Activation Regions and Hyperplane Arrangements

Activation regions in depth 1 ReLU nets are given by hyperplane arrangements in $\mathbb{R}^{n_{\text{in}}}$ (see [27]). Indeed, if $\mathcal{N}$ is a ReLU net with one hidden layer, then the sets $H_z(\theta)$ from (2) are simply hyperplanes, giving the well-known observation that the activation regions in a depth 1 ReLU net are the connected components of $\mathbb{R}^{n_{\text{in}}}$ with the hyperplanes $H_z(\theta)$ removed. The study of regions induced by hyperplane arrangements in $\mathbb{R}^n$ is a classical subject in combinatorics [27]. A basic result is that for hyperplanes in general position (e.g. chosen at random), the total number of connected components coming from an arrangement of $m$ hyperplanes in $\mathbb{R}^n$ is constant:

$$\#\text{connected components} \;=\; \sum_{i=0}^{n} \binom{m}{i} \;\simeq\; \begin{cases} \frac{m^n}{n!}, & m \gg n \\ 2^m, & m \leq n \end{cases}. \tag{4}$$

Hence, for random $w_j, b_j$ drawn from any reasonable distributions the number of activation regions in a ReLU net with input dimension $n_{\text{in}}$ and one hidden layer of size $m$ is given by (4). The situation is more subtle for deeper networks. By Lemma 2, activation regions are connected components for an arrangement of "bent" hyperplanes $H_z(\theta)$ from (2), which are only locally described by hyperplanes. To understand their structure more carefully, fix a ReLU net $\mathcal{N}$ with $d$ hidden layers and a vector $\theta$ of trainable parameters for $\mathcal{N}$. Write $\mathcal{N}_j$ for the network obtained by keeping only the first $j$ layers of $\mathcal{N}$ and $\theta_j$ for the corresponding parameter vector. The following lemma makes precise the observation that the hyperplane $H_z(\theta)$ can only bend only when it meets a bent hyperplane $H_{\widehat{z}}(\theta)$ corresponding to some neuron $\widehat{z}$ in an earlier layer.

**Lemma 4** ($H_z(\theta)$ as Bent Hyperplanes). *Except on a set of $\theta \in \mathbb{R}^{\#\text{params}}$ of measure $0$ with respect to Lebesgue measure, the sets $H_z(\theta_1)$ corresponding to neurons from the first hidden layer are hyperplanes in $\mathbb{R}^{n_{\text{in}}}$. Moreover, fix $2 \leq j \leq d$. Then, for each neuron $z$ in layer $j$, the set $H_z(\theta_j)$ coincides with a single hyperplane in the interior of each activation region of $\mathcal{N}_{j-1}$.*

Lemma 4, which follows immediately from the proof of Lemma 7 in Appendix B.1, ensures that in a small ball near any point that does not belong to $\bigcup_z H_z(\theta)$, the collection of bent hyperplanes $H_z(\theta)$ look like an ordinary hyperplane arrangement. Globally, however, $H_z(\theta)$ can define many more regions than ordinary hyperplane arrangements. This reflects the fact that deep ReLU nets may have many more activation regions than shallow networks with the same number of neurons.

Despite their different extremal behaviors, we show in Theorem 5 that the average number of activation regions in a random ReLU net enjoys depth-independent upper bounds at initialization. We show experimentally that this holds throughout training as well (see §4). On the other hand, although we do not prove this here, we believe that the effect of depth can be seen through the *fluctuations* (e.g. the variance), rather than the mean, of the number of activation regions. For instance, for depth 1 ReLU nets, the variance is 0 since for a generic configuration of weights/biases, the number of activation regions is constant (see (4)). The variance is strictly positive, however, for deeper networks.

# 3 Main Result

## 3.1 Formal Statement

Theorem 5 gives upper bounds on the average number of activation regions per unit volume of input space for a feed-forward ReLU net with random weights/biases. Note that it applies even to highly correlated weight/bias distributions and hence holds throughout training. Also note that although we require no tied weights, there are no further constraints on the connectivity between adjacent layers.

**Theorem 5** (Counting Activation Regions). *Let $\mathcal{N}$ be a feed-forward ReLU network with no tied weights, input dimension $n_{\mathrm{in}}$, output dimension $1$, and random weights/biases satisfying:*

1. *The distribution of all weights has a density with respect to Lebesgue measure on $\mathbb{R}^{\#\mathrm{weights}}$.*

2. *Every collection of biases has a density with respect to Lebesgue measure conditional on the values of all weights and other biases (for identically zero biases, see Appendix D).*

3. *There exists $C_{\mathrm{grad}} > 0$ so that for every neuron $z$ and each $m \geq 1$, we have*
$$\sup_{x \in \mathbb{R}^{n_{\mathrm{in}}}} \mathbb{E}\left[\|\nabla z(x)\|^m\right] \leq C_{\mathrm{grad}}^m.$$

4. *There exists $C_{\mathrm{bias}} > 0$ so that for any neurons $z_1, \ldots, z_k$, the conditional distribution of the biases $\rho_{b_{z_1}, \ldots, b_{z_k}}$ of these neurons given all the other weights and biases in $\mathcal{N}$ satisfies*
$$\sup_{b_1, \ldots, b_k \in \mathbb{R}} \rho_{b_{z_1}, \ldots, b_{z_k}}(b_1, \ldots, b_k) \leq C_{\mathrm{bias}}^k.$$

*Then, there exists $\delta_0, T > 0$ depending on $C_{\mathrm{grad}}, C_{\mathrm{bias}}$ with the following property. Suppose that $\delta > \delta_0$. Then, for all cubes $\mathcal{C}$ with side length $\delta$, we have*

$$\frac{\mathbb{E}\left[\#\text{non-empty activation regions of } \mathcal{N} \text{ in } \mathcal{C}\right]}{\mathrm{vol}(\mathcal{C})} \leq \begin{cases} (T\#\mathrm{neurons})^{n_{\mathrm{in}}}/n_{\mathrm{in}}! & \#\mathrm{neurons} \geq n_{\mathrm{in}} \\ 2^{\#\mathrm{neurons}} & \#\mathrm{neurons} \leq n_{\mathrm{in}} \end{cases}.$$
(5)

*Here, the average is with respect to the distribution of weights and biases in $\mathcal{N}$.*

**Remark 1.** The heuristic of §3.3 suggests the average number of activation patterns in $\mathcal{N}$ over all of $\mathbb{R}^{n_{\mathrm{in}}}$ is at most $(\#\mathrm{neurons})^{n_{\mathrm{in}}}/n_{\mathrm{in}}!$, its value for depth 1 networks (see (4)). This is confirmed in our experiments (see Figures 3-6).

We state and prove a generalization of Theorem 5 in Appendix C. Note that by Theorem 1 (and Proposition 2) in [12], Condition 3 is automatically satisfied by a fully connected depth $d$ ReLU net $\mathcal{N}$ with independent weights and biases whose marginals are symmetric around 0 and satisfy $\mathrm{Var}[\mathrm{weights}] = 2/\mathrm{fan\text{-}in}$ with the constant $C_{\mathrm{grad}}$ in 3 depending only on an upper bound for the sum $\sum_{j=1}^{d} 1/n_j$ of the reciprocals of the hidden layer widths of $\mathcal{N}$. For example, if the layers of $\mathcal{N}$ have constant width $n$, then $C_{\mathrm{grad}}$ depends on the depth and width only via the aspect ratio $d/n$ of $\mathcal{N}$, which is small for wide networks. Also, at initialization when all biases are independent, the constant $C_{\mathrm{bias}}$ can be taken simply to be the maximum of the density of the bias distribution.

Below are two heuristics for the second (5). First, in §3.2 we derive the upper bound (5) via an intuitive geometric argument. Then in §3.3, we explain why, at initialization, we expect the upper bounds (5) to have matching, depth-independent, lower bounds (to leading order in the number of neurons). This suggests that the average *total* number of activation regions at initialization should be the same for any two ReLU nets with the same number of neurons (see (4) and Figure 3).

## 3.2 Geometric Intuition

We give an intuitive explanation for the upper bounds in Theorem 5, beginning with the simplest case of a ReLU net $\mathcal{N}$ with $n_{\text{in}} = 1$. Activation regions for $\mathcal{N}$ are intervals, and at an endpoint $x$ of such an interval the pre-activation of some neuron $z$ in $\mathcal{N}$ equals its bias: i.e. $z(x) = b_z$. Thus,

$$\#\text{activation regions of } \mathcal{N} \text{ in } [a, b] \ \leq \ 1 + \sum_{\text{neurons } z} \#\{x \in [a, b] \mid z(x) = b_z\}.$$

Geometrically, the number of solutions to $z(x) = b_z$ for inputs $x \in I$ is the number of times the horizontal line $y = b_z$ intersects the graph $y = z(x)$ over $x \in I$. A large number of intersections at a given bias $b_z$ may only occur if the graph of $z(x)$ has many oscillations around that level. Hence, since $b_z$ is random, the graph of $z(x)$ must oscillate many times over a large range on the $y$ axis. This can happen only if the total variation $\int_{x \in I} |z'(x)|$ of $z(x)$ over $I$ is large. Thus, if $|z'(x)|$ is typically of moderate size, we expect only $O(1)$ solutions to $z(x) = b_z$ per unit input length, suggesting

$$\mathbb{E}\left[\#\text{activation regions of } \mathcal{N} \text{ in } [a, b]\right] \ = \ O\left((b - a) \cdot \#\text{neurons}\right),$$

in accordance with Theorem 5 (cf. Theorems 1,3 in [14]). When $n_{\text{in}} > 1$, the preceding argument, shows that density of 1-dimensional regions per unit length along any 1-dimensional line segment in input space is bounded above by the number of neurons in $\mathcal{N}$. A unit-counting argument therefore suggests that the density of $n_{\text{in}}$-dimensional regions per unit $n_{\text{in}}$-dimensional volume is bounded above by #neurons raised to the input dimension, which is precisely the upper bound in Theorem 5 in the non-trivial regime where $\#\text{neurons} \gg n_{\text{in}}$.

## 3.3 Is Theorem 5 Sharp?

Theorem 5 shows that, on average, depth does not increase the *local density* of activation regions. We give here an intuitive explanation of why this should be the case in wide networks on any fixed subset of input space $\mathbb{R}^{n_{\text{in}}}$. Consider a ReLU net $\mathcal{N}$ with random weights/biases, and fix a layer index $\ell \geq 1$. Note that the map $x \mapsto x^{(\ell-1)}$ from inputs $x$ to the post-activations of layer $\ell - 1$ is itself a ReLU net. Note also that in wide networks, the gradients $\nabla z(x)$ for different neurons in the same layer are only weakly correlated (cf. e.g. [17]). Hence, for the purpose of this heuristic, we will assume that the bent hyperplanes $H_z(\theta)$ for neurons $z$ in layer $\ell$ are independent. Consider an activation region $\mathcal{R}$ for $x^{(\ell-1)}(x)$. By definition, in the interior of $\mathcal{R}$, the gradient $\nabla z(x)$ for neurons $z$ in layer $\ell$ are constant and hence the corresponding bent hyperplane from (2) inside $\mathcal{R}$ is the hyperplane $\{x \in \mathcal{R} \mid \langle \nabla z, x \rangle = b_z\}$. This in keeping with Lemma 4. The $2/\text{fan-in}$ weight normalization ensures that for each $x$

$$\mathbb{E}\left[\partial_{x_i} \partial_{x_j} z(x)\right] \ = \ 2 \cdot \delta_{i,j} \quad \Rightarrow \quad \text{Cov}[\nabla z(x)] \ = \ 2\,\text{Id}.$$

See, for example, equation (17) in [11]. Thus, the covariance matrix of the normal vectors $\nabla z$ of the hyperplanes $H_z(\theta) \cap \mathcal{R}$ for neurons in layer $\ell$ are *independent of $\ell$*! This suggests that, per neuron, the average contribution to the number of activation regions is the same in every layer. In particular, deep and shallow ReLU nets with the same number of neurons should have the same average number of activation regions (see (4), Remark 1, and Figures 3-6).

## 4 Maximizing the Number of Activation Regions

While we have seen in Figure 3 that the number of regions does not strongly increase during training on a simple task, such experiments leave open the possibility that the number of regions would go up markedly if the task were more complicated. Will the number of regions grow to achieve the theoretical upper bound (exponential in the depth) if the task is designed so that having more regions is advantageous? We now investigate this possibility. See Appendix A for experimental details.

### 4.1 Memorization

Memorization tasks on large datasets require learning highly oscillatory functions with large numbers of activation regions. Inspired by the work of Arpit et. al. in [1], we train on several tasks interpolating between memorization and generalization (see Figure 4) in a certain fraction of MNIST labels have been randomized. We find that the maximum number of activation regions learned does increase with the amount of noise to be memorized, but only slightly. In no case does the number of activation regions change by more than a small constant factor from its initial value. Next, we train a network to memorize binary labels for random 2D points (see Figure 5). Again, the number of activation regions after training increases slightly with increasing memorization, until the task becomes too hard for the network and training fails altogether. Varying the learning rate yields similar results (see Figure 6(a)), suggesting the small increase in activation regions is probably not a result of hyperparameter choice.

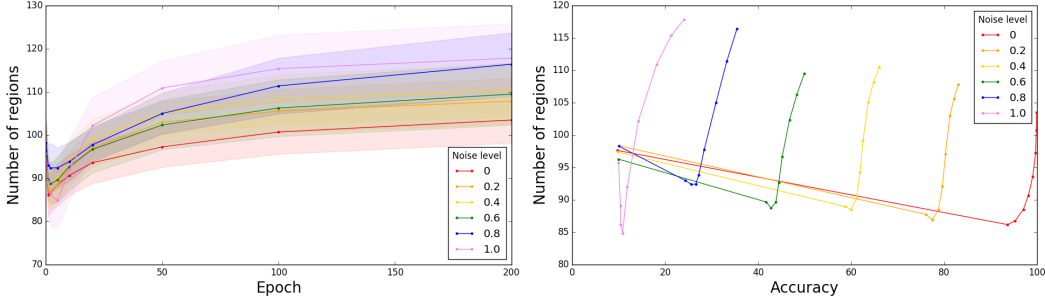

Figure 4: Depth 3, width 32 network trained on MNIST with varying levels of label corruption. Activation regions are counted along lines through input space (lines are selected to pass through both the origin and randomly selected MNIST examples), with counts averaged across 100 such lines. Theorem 5 and [14] predict the expected number of regions should be approximately the number of neurons (in this case, 96). Left: average number of regions plotted against epoch. Curves are averaged over 40 independent training runs, with standard deviations shown. Right: average number of regions plotted against average training accuracy. Throughout training the number of regions is well-predicted by our result. There are slightly, but not exponentially, more regions when memorizing more datapoints. See Appendix A for more details.

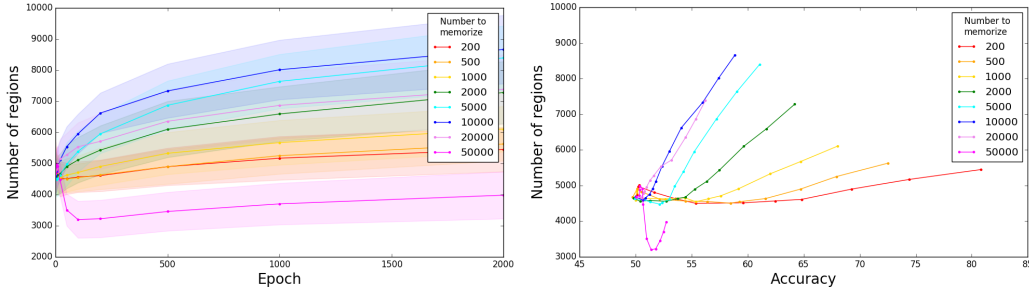

Figure 5: Depth 3, width 32 fully connected ReLU net trained for 2000 epochs to memorize random 2D points with binary labels. The number of regions predicted by Theorem 5 for such a network is $96^2/2! = 4608$. Left: number of regions plotted against epoch. Curves are averaged over 40 independent training runs, with standard deviations shown. Right: #regions plotted against training accuracy. The number of regions increased during training, and increased more for greater amounts of memorization. The exception was for the maximum amount of memorization, where the network essentially failed to learn, perhaps because of insufficient capacity. See Appendix A for more details.

## 4.2 The Effect of Initialization

We explore here whether varying the scale of biases and weights at initialization affects the number of activation regions in a ReLU net. Note that scaling the biases changes the maximum density of the bias, and thus affects the upper bound on the density of activation regions given in Theorem 5 by increasing $T_{\text{bias}}$. Larger, more diffuse biases reduce the upper bound, while smaller, more tightly concentrated biases increase it. However, Theorem 5 counts only the *local* rather than *global* number of regions. The latter are independent of scaling the biases:

**Lemma 6.** *Let $\mathcal{N}$ be a deep ReLU network, and for $c > 0$ let $\mathcal{N}_c^{bias}$ be the network obtained by multiplying all biases in $\mathcal{N}$ by $c$. Then, $\mathcal{N}(x) = \mathcal{N}_c^{bias}(cx)/c$. Rescaling all biases by the same constant therefore does not change the total number of activation regions.*

In the extreme case of biases initialized to zero, Theorem 5 does not apply. However, as we explain in Appendix D, zero biases only create fewer activation regions (see Figure 7). We now consider changing the scale of weights at initialization. In [23], it was suggested that initializing the weights of a network with greater variance should increase the number of activation regions. Likewise, the upper bound in Theorem 5 on the density of activation regions increases as gradient norms increase, and it has been shown that increased weight variance increases gradient norms [12]. However, this is again a property of the local, rather than global, number of regions.

Indeed, for a network $\mathcal{N}$ of depth $d$, write $\mathcal{N}_c^{\text{weight}}$ for the network obtained from $\mathcal{N}$ by multiplying all its weights by $c$, and let $\mathcal{N}_{1/c*}^{\text{bias}}$ be obtained from $\mathcal{N}$ by dividing the biases in the $k$th layer by $c^k$.

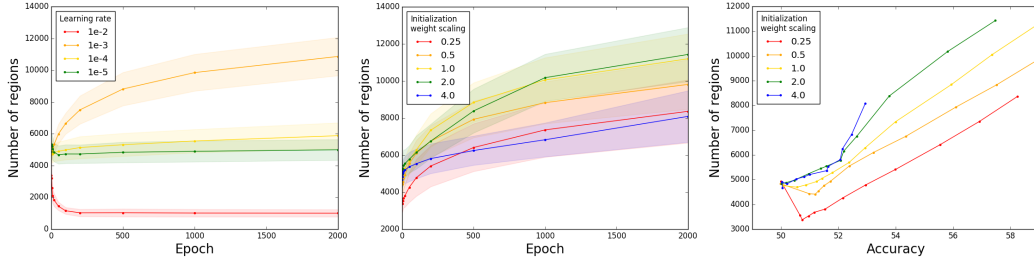

Figure 6: Depth 3, width 32 network trained to memorize 5000 random 2D points with independent binary labels, for various learning rates and weight scales at initialization. All networks start with $\approx 4608$ regions, as predicted by Theorem 5. Left: None of the learning rates gives a number of regions larger than a small constant times the initial value. Learning rate $10^{-3}$, which gives the maximum number of regions, is the learning rate in all other experiments, while $10^{-2}$ is too large and causes learning to fail. Center: Different weight scales at initialization do not strongly affect the number of regions. All weight scales are given relative to variance $2/\text{fan-in}$. Right: For a given accuracy, the number of regions learned grows with the weight scale at initialization. However, poor initialization impedes high accuracy. See Appendix A for details.

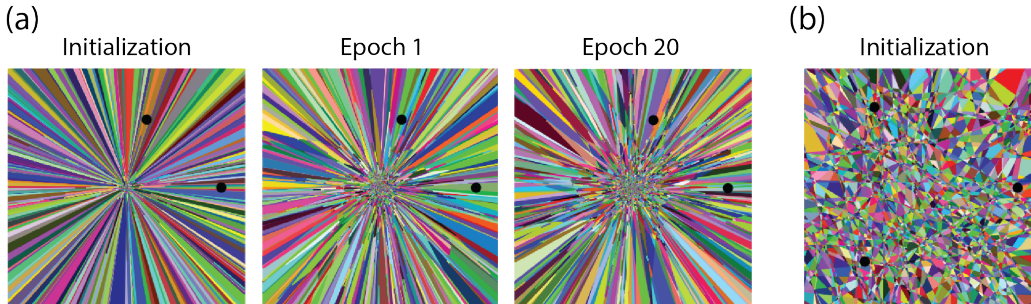

Figure 7: Activation regions within input space, for a network of depth 3 and width 64 training on MNIST. (a) Cross-section through the origin, shown at initialization, after one epoch, and after twenty epochs. The plane is chosen to pass through two sample points from MNIST, shown as black dots. (b) Cross-section not through the origin, shown at initialization. The plane is chosen to pass through three sample points from MNIST. For discussion of activation regions at zero bias, see Appendix D.

A scaling argument shows that $\mathcal{N}_c^{\text{weight}}(x) = c^d \mathcal{N}_{1/c*}^{\text{bias}}(x)$. We therefore conclude that the activation regions of $\mathcal{N}_c^{\text{weight}}$ and $\mathcal{N}_{1/c*}^{\text{bias}}$ are the same. Thus, scaling the weights uniformly is equivalent to scaling the biases differently for every layer. We have seen from Lemma 6 that scaling the biases uniformly by any amount does not affect the global number of activation regions. Therefore, it makes sense (though we do not prove it) that scaling the weights uniformly should approximately preserve the global number of activation regions. We test this intuition empirically by attempting to memorize points randomly drawn from a 2D input space with arbitrary binary labels for various initializations (see Figure 6). We find that neither at initialization nor during training is the number of activation regions strongly dependent on the weight scaling used for initialization.

## 5  Conclusion

We have presented theoretical and empirical evidence that the number of activation regions learned in practice by a ReLU network is far from the maximum possible and depends mainly on the number of neurons in the network, rather than its depth. This surprising result implies that, at least when network gradients and biases are well-behaved (see conditions 3,4 in the statement of Theorem 5), the partition of input space learned by a deep ReLU network is not significantly more complex than that of a shallow network with the same number of neurons. We found that this is true even after training on memorization-based tasks, in which we expect a large number of regions to be advantageous for fitting many randomly labeled inputs. Our results are stated for ReLU nets with no tied weights and biases (and arbitrary connectivity). We believe that analogous results and proofs hold for residual and convolutional networks but have not verified the technical details.

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
