[Supplementary Material]

# A  Experimental Design

We run several experiments that involve calculating the activation regions intersecting a 1D or 2D subset of input space. In order to compute these regions, we add neurons of the network one by one from the first to last hidden layer, observing how each neuron cuts existing regions. Determining whether a region is cut by a neuron involves identifying whether the corresponding linear function on that region has zeros within the region. This can be solved easily by identifying whether all vertices of the region have the same sign (region is not cut) or some two of the vertices have different signs (region is cut). Thus, our procedure is to maintain a list of regions and the linear functions defined on them, then for each newly added neuron identify on which regions its preactivation vanishes and replace these regions by the resulting split regions. Note that this procedure also works in three dimensions and higher, but becomes slower as the number of regions in higher dimensions grows like the number of neurons to the dimension, as shown in Theorem 5.

Unless otherwise specified, all experiments involving training a network were performed using an Adam optimizer, with learning rate $10^{-3}$ and batch size 128. Networks are, unless otherwise specified, initialized with i.i.d. normal weights with variance $2/\text{fan-in}$ (for justification of this initialization with ReLU networks, see [13, 15]) and i.i.d. normal biases with variance $10^{-6}$.

# B  Proofs of Various Lemmas

## B.1  Statement and Proof of Lemma 1 for General Piecewise Linear Activations

We begin by formulating Lemma 1 for a general continuous piecewise linear function $\varphi : \mathbb{R} \to \mathbb{R}$. For such a $\phi$, there exists a non-negative integer $T \geq 0$, as well as

$$-\infty = \xi_0 < \xi_1 < \cdots < \xi_T, \qquad p_0, q_0, \ldots, p_T, q_T \in \mathbb{R}, \quad \text{with } p_i \neq p_{i+1} \text{ for } i = 0, \ldots, T-1$$

so that

$$\varphi(t) = p_i t + q_i, \quad t \in (\xi_i, \xi_{i+1}].$$

**Definition 3.** *Let $\mathcal{N}$ be a network with input dimension $n_{\text{in}}$ and non-linearity $\phi$. An activation pattern for $\mathcal{N}$ assigns to each neuron an element of the alphabet $\{0, \ldots, T\}$:*

$$\mathcal{A} := \{a_z, \ z \text{ a neuron in } \mathcal{N}\} \in \{0, \ldots, T\}^{\#\text{neurons}}.$$

*Fix $\theta$, a vector of trainable parameters in $\mathcal{N}$, and an activation pattern $\mathcal{A}$. The activation region corresponding to $\mathcal{A}, \theta$ is*

$$\mathcal{R}(\mathcal{A}; \theta) := \{x \in \mathbb{R}^{n_{\text{in}}} \mid z(x) - b_z \in (\xi_{a_z}, \xi_{a_z+1}), \quad z \text{ a neuron in } \mathcal{N}\},$$

*where the pre-activation of a neuron $z$ is $z(x; \theta)$, its bias is $b_z$, and its post-activation is therefore $\varphi(z(x; \theta) - b_z)$. Finally, the activation regions of $\mathcal{N}$ at $\theta$ is the collection of all non-empty activation regions $\mathcal{R}(\mathcal{A}, \theta)$.*

We will prove the following generalization of Lemma 8.

**Lemma 7** (Activation Regions are Convex). *Let $\mathcal{N}$ be a network with non-linearity $\varphi$, and let*

$$\mathcal{A} = \{a_{j,z}, \ z \text{ a neuron in } \mathcal{N}\} \in \{0, \ldots, T\}^{\#\text{neurons}}$$

*be any activation pattern. Then, for any vector $\theta$ of trainable parameters for $\mathcal{N}$, the region $\mathcal{R}(\mathcal{A}; \theta)$ is convex.*

*Proof.* Write $d$ for the depth of $\mathcal{N}$ and note that, by definition,

$$\mathcal{R}(\mathcal{A}; \theta) = \bigcap_{\ell=1}^{d} \bigcap_{\substack{\text{neurons } z \\ \text{in layer } \ell}} \mathcal{R}_z(\mathcal{A}; \theta), \quad \mathcal{R}_z(\mathcal{A}; \theta) := \{x \in \mathbb{R}^{n_{\text{in}}} \mid z(x) - b_z \in (-\xi_{a_z}, \xi_{a_z+1})\}.$$

We will show that $\mathcal{R}(\mathcal{A}, \theta)$ is convex by induction on $d$. For the base case, note that given $s < t \in [-\infty, \infty]$, for every $w \in \mathbb{R}^{n_{\text{in}}}$, $b \in \mathbb{R}$

$$\{x \in \mathbb{R}^{n_{\text{in}}} \mid w \cdot x - b \in (s, t)\}$$

is convex. Hence, the intersection of any number of such sets is convex as well. Note that when $d = 1$, $R_z(\mathcal{A}; \theta)$ is of this form every $z$ proves the base case. For the inductive case, suppose we

have shown the claim for some $d \geq 1$. For inputs $x$ in the $\bigcap_{\ell=1}^{d} \bigcap_{\substack{\text{neurons } z \\ \text{in layer } \ell}} \mathcal{R}_z(\mathcal{A}; \theta)$, there exists, for every neuron $z$ in layer $d + 1$ a vector $w_z \in \mathbb{R}^{n_{\text{in}}}$ and a scalar $b_z \in \mathbb{R}$ so that

$$z(x) = w_z \cdot x - b.$$

Therefore,

$$\bigcap_{\substack{\ell=1 \\ \text{neurons } z \\ \text{in layer } \ell}}^{d} \mathcal{R}_z(\mathcal{A}; \theta) \ \cap \ R_z(\mathcal{A}; \theta) \ = \ \{x \in \mathcal{R}_d \mid w_z \cdot x - b \in (\xi_{a_z}, \xi_{a_z+1})\}$$

is the intersection of two convex sets and is therefore convex. Taking the intersection over all $z$ in layer $d + 1$ completes the proof. □

## B.2 Proof of Lemma 2

We claim the following general fact. Suppose $X$ is a topological space and $f_j : X \to \mathbb{R}$ are continuous, $j = 1, \ldots, J$. Then, on every connected component of $X \backslash \bigcup_j f_j^{-1}(0)$, the sign of $f$ is constant. Indeed, consider a connected component $X_\alpha$. Since $f_j$ are never $0$ on $X_\alpha$ by construction, we have $f_j(X_\alpha) \subseteq (-\infty, 0) \cup (0, \infty)$. But the image under a continuous map of a connected set is connected. Hence, for each $j$ $f_j(X_\alpha) \subseteq (-\infty, 0)$ or $f_j(X_\alpha) \subseteq (0, \infty)$, and the claim follows.

Turning to Lemma 2, let $\mathcal{N}$ be a ReLU net with input dimension $n_{\text{in}}$, and fix a vector of trainable parameters $\theta$ for $\mathcal{N}$. The claim above shows that on every connected component of $\mathbb{R}^{n_{\text{in}}} \setminus \bigcup_{\text{neurons } z} H_z(\theta)$ the functions $z(x) - b_z$ have a definite sign for all neurons $z$. Thus, every connected component is contained in some activation region $\mathcal{R}(\mathcal{A}; \theta)$. Finally, by construction,

$$\mathcal{R}(\mathcal{A}; \theta) \ \subset \ \mathbb{R}^{n_{\text{in}}} \setminus \bigcup_{\text{neurons } z} H_z(\theta)$$

and, by Lemma 1, $\mathcal{R}(\mathcal{A}; \theta)$ is convex and hence connected. Therefore it is equal to the connected component we started with. □

## B.3 Proof of Lemma 3

Let $\mathcal{N}$ be a ReLU net, and fix a vector $\theta$ of its trainable parameters. Let us first check that

$$\#\text{linear regions}\,(\mathcal{N}, \theta) \ \leq \ \#\text{activation regions}\,(\mathcal{N}, \theta). \tag{6}$$

We will use the following simple fact: if $X, Y$ are subsets of a topological space $T$, then $X \subset Y$ implies that every connected component of $X$ is the subset of some connected component of $Y$. Indeed, if $X_\alpha$ is a connected component of $X$, then it is a connected subjset of $Y$ and hence is contained in a unique connected component of $Y$.

This fact shows that the cardinality of the set of connected components of $Y$ is bounded above by the cardinality of the set of connected components for $X$. Using Lemma 2, the inequality (6) therefore reduces to showing that

$$\mathcal{B}_{\mathcal{N}}(\theta) \ \subseteq \ \bigcup_{\text{neurons } z} H_z(\theta), \quad H_z(\theta) = \{x \in \mathbb{R}^{n_{\text{in}}} \mid z(x) = b_z\}. \tag{7}$$

Fix any $x$ in the complement of the right hand side. By definition, we have

$$|z(x) - b_z| > 0, \qquad \forall \text{ neurons } z.$$

The functions $z(x)$ are continuous. Hence, in a small neighborhood of $x$, the estimate in the previous line holds in a small neighborhood of $x$. Thus, in an open neighborhood of $x$, the collection of neurons that are on and off are constant. The inclusion (7) now follows by observing that if for all $y$ in an open neighborhood $U$ of $x \in \mathbb{R}^{n_{\text{in}}}$, the sets

$$\mathcal{A}_{y,\pm}(\theta) \ := \ \{\text{neurons } z \mid \ \pm (z(y; \theta) - b_z) > 0\}$$

are constant and

$$\mathcal{A}_{y,0}(\theta) \ := \ \{\text{neurons } z \mid z(y; \theta) = b_z\} \ = \ \emptyset,$$

then $\mathcal{N}$ restricted to $U$ is given by a single linear function and hence has a continuous gradient $\nabla \mathcal{N}(\cdot\,; \theta)$ on $U$.

It remains to check that the closure of every linear region of $\mathcal{N}$ at $\theta$ is the closure of the union of some activation regions of $\mathcal{N}$ at $\theta$. Note that, except on a set of $\theta \in \mathbb{R}^{\text{params}}$ with Lebesgue measure $0$, both $\bigcup_z H_z(\theta)$ and $\mathcal{B}_{\mathcal{N}}(\theta)$ are co-dimension 1 piecewise linear submanifolds of $\mathbb{R}^{n_{\text{in}}}$ with finitely many pieces. Thus, $\mathbb{R}^{n_{\text{in}}} \backslash \bigcup_z H_z(\theta)$ is open and dense in $\mathbb{R}^{n_{\text{in}}} \backslash \mathcal{B}_{\mathcal{N}}(\theta)$. And now we appeal to a general topological fact: if $X \subset Y$ are subsets of a topological space $T$ and $X$ is open and dense in $Y$, then the closure in $T$ of every connected component of $Y$ is the closure of the union of some connected components of $X$. Indeed, consider a connected component $Y_\beta$ of $Y$. Then $X \cap Y_\beta$ is open and dense in $Y_\beta$. On the other hand, $X \cap Y_\beta$ is the union of connected components of $X_\alpha$ of $X$. Thus, the closure of this union is the closure of $X \cap Y_\beta$, namely the closure of $Y_\beta$. $\qquad\square$

### B.4 Distinguishability of Activation Regions

In addition to being convex, activation regions for a ReLU net $\mathcal{N}$ generically correspond to different linear functions:

**Lemma 8** (Activation Regions are Distinguishable). *Let $\mathcal{N}$ be a ReLU net, and let*

$$\mathcal{A}_j = \{a_{j,z}, \ z \text{ a neuron in } \mathcal{N}\} \in \{-1,1\}^{\#\text{neurons}} \quad j = 1,2$$

*be two activation patterns with $\mathcal{R}(\mathcal{A}_j; \theta) \neq \emptyset$. Suppose also that, for every layer and each $j = 1,2$ there exists a neuron $z_j$ with $a_{j,z_j} = 1$. Then, except on a measure zero set of $\theta$ with respect to Lebesgue measure in the parameter space $\mathbb{R}^{\#\text{params}}$, the gradient $\nabla \mathcal{N}(x; \theta)$ is different for $x$ in the interiors of $\mathcal{R}(\mathcal{A}_j; \theta)$.*

*Proof.* Fix an activation pattern $\mathcal{A}$ for a depth $d$ ReLU network $\mathcal{N}$ with $\mathcal{R}(\mathcal{A}; \theta) \neq \emptyset$. Fix $x \in \mathcal{R}(\mathcal{A}; \theta)$. We will use the following well-known formula:

$$\nabla \mathcal{N}(x; \theta) \ = \ \sum_{\text{paths } \gamma} \mathbf{1}_{\{\gamma \text{ open at } x\}} \prod_{i=1}^{d} w_\gamma^{(j)},$$

where the sum is over all paths in the computational graph of $\mathcal{N}$, a path is open at $x$ if every neuron $z$ in $\gamma$ satisfies $z(x) - b_z > 0$, and $w_\gamma^{(j)}$ is the weight on the edge of $\gamma$ between layers $j-1$ and $j$. If there exist two different, non-empty activation regions corresponding to activation patterns $\mathcal{A}$ for which there is at least one open path through the network on which $\nabla N(\,\cdot\,; \theta)$ has the same value on , then there exists $a_\gamma \in \{\pm 1\}$ and a non-empty collection $\Gamma$ of paths so that

$$\sum_{\gamma \in \Gamma} a_\gamma \prod_{i=1}^{d} w_\gamma^{(j)} \ = \ 0. \tag{8}$$

The zero set of any such polynomial (since $\Gamma \neq \emptyset$) is a co-dimension 1 variety in $\mathbb{R}^{\#\text{weights}}$. Since there are only finitely many (in fact $2^{\#\text{paths}}$) such polynomials, the set of $\theta$ for which (8) can occur has measure 0 with respect to the Lebesgue measure on $\mathbb{R}^{\#\text{weights}}$, as claimed. $\qquad\square$

## C Statement and Proof of a Generalization of Theorem 5

We begin by stating a generalization of Theorem 5 to what we term partial activation regions. Let us write

$$\text{sgn}(t) := \begin{cases} 1, & t > 0 \\ 0, & t = 0 \\ -1, & t < 0 \end{cases}.$$

**Definition 4** (Partial Activation Regions). *Let $\mathcal{N}$ be a ReLU net with input dimension $n_{\text{in}}$. Fix a non-negative integer $k$. A $k$-partial activation pattern for $\mathcal{N}$ is an assignment to each neuron of a sign of $-1, 0, 1$, with exactly $k$ neurons being assigned a $0$:*

$$\mathcal{A} := \{a_z, \ z \text{ a neuron in } \mathcal{N}\} \in \{-1,0,1\}^{\#\text{neurons}}, \quad \#\{z \mid a_z = 0\} = k.$$

*Fix $\theta$, a vector of trainable parameters in $\mathcal{N}$, and a $k$-partial activation pattern $\mathcal{A}$. The $k$-partial activation region corresponding to $\mathcal{A}, \theta$ is*

$$\mathcal{R}(\mathcal{A}; \theta) := \{x \in \mathbb{R}^{n_{\text{in}}} \mid \text{sgn}(z(x; \theta) - b_z) = a_z, \quad z \text{ a neuron in } \mathcal{N}\},$$

*where the pre-activation of a neuron $z$ is $z(x; \theta)$, its bias is $b_z$, its post-activation is therefore $\max\{0, z(x; \theta) - b_z\}$. Finally, the activation regions of $\mathcal{N}$ at $\theta$ is the collection of all non-empty activation regions $\mathcal{R}(\mathcal{A}, \theta)$.*

The same argument as the proof of Lemma 7 yields the following result:

**Lemma 9.** *Let $\mathcal{N}$ be a ReLU net. Fix a non-negative integer $r$ and let $\mathcal{A}$ be a $r$-partial activation pattern for $\mathcal{N}$. For every vector $\theta$ of trainable parameters for $\mathcal{N}$, the $r$-partial activation region $\mathcal{R}(\mathcal{A};\theta)$ is convex.*

We will prove the following generalization of Theorem 5.

**Theorem 10.** *Let $\mathcal{N}$ be a feed-forward ReLU network with no tied weights, input dimension $n_{\mathrm{in}}$ and output dimension $1$. Suppose that the weights and biases $\mathcal{N}$ is random and satisfies:*

1. *The distribution of all weights has a density with respect to Lebesgue measure on $\mathbb{R}^{\#\mathrm{weights}}$.*

2. *Every collection of biases has a density with respect to Lebesgue measure conditional on the values of all weights and other biases (for identically zero biases, see Appendix D).*

3. *There exists $C_{\mathrm{grad}} > 0$ so that for every neuron $z$ and each $m \geq 1$, we have*
$$\sup_{x \in \mathbb{R}^{n_{\mathrm{in}}}} \mathbb{E}\left[\|\nabla z(x)\|^m\right] \leq C_{\mathrm{grad}}^m.$$

4. *There exists $C_{\mathrm{bias}} > 0$ so that for any neurons $z_1, \ldots, z_k$, the conditional distribution of the biases $\rho_{b_{z_1}, \ldots, b_{z_k}}$ of these neurons given all the other weights and biases in $\mathcal{N}$ satisfies*
$$\sup_{b_1, \ldots, b_k \in \mathbb{R}} \rho_{b_{z_1}, \ldots, b_{z_k}}(b_1, \ldots, b_k) \leq C_{\mathrm{bias}}^k.$$

*Fix $r \in \{0, \ldots, n_{\mathrm{in}}\}$. Then, there exists $\delta_0, T > 0$ depending on $C_{\mathrm{grad}}, C_{\mathrm{bias}}$ with the following property. Suppose that $\delta > \delta_0$. Then, for all cubes $\mathcal{C}$ with side length $\delta$, we have*

$$\frac{\mathbb{E}\left[\#r - \text{partial activation regions of } \mathcal{N} \text{ in } \mathcal{C}\right]}{\mathrm{vol}(\mathcal{C})} \leq \begin{cases} \frac{(T\#\mathrm{neurons})^{n_{\mathrm{in}}}}{2^r n_{\mathrm{in}}!}, & \#\mathrm{neurons} \geq n_{\mathrm{in}} \\ 2^{\#\mathrm{neurons}}, & \#\mathrm{neurons} \leq n_{\mathrm{in}} \end{cases}. \quad (9)$$

*Here, the average is with respect to the distribution of weights and biases in $\mathcal{N}$.*

*Proof.* Fix a ReLU net $\mathcal{N}$ with input dimension $n_{\mathrm{in}}$ and a non-negative integer $r$. Since the number of distinct $r$-partial activation patterns in $\mathcal{N}$ is at most $\binom{\#\mathrm{neurons}}{r} 2^{\#\mathrm{neurons}-r}$, Theorem 5 only requires proof when $\#\mathrm{neurons} \geq n_{\mathrm{in}}$. For this, for any vector of trainable parameters $\theta$ define

$$X_{\mathcal{N},k}(\theta) := \bigcup_{\substack{\text{collections } I \text{ of } k \\ \text{distinct neurons in } \mathcal{N}}} \left[\bigcap_{z \in I} H_z(\theta) \cap \bigcap_{z \notin I} H_z(\theta)^c\right].$$

In words, $X_{\mathcal{N},k}$ is the collection of inputs $x \in \mathbb{R}^{n_{\mathrm{in}}}$ for which there exist exactly $k$ neurons $z$ so that $x$ solves $z(x) = b_z$. We record for later use the following fact.

**Lemma 11.** *With probability $1$ over the space of $\theta$'s, for any $x \in X_{\mathcal{N},k}$ there exists $\varepsilon > 0$ (depending on $x$ and $\theta$) so that the set $X_{\mathcal{N},k}$ intersected with the $\varepsilon$ ball $B_\varepsilon(x)$ coincides with a hyperplane of dimension $n_{\mathrm{in}} - k$.*

*Proof.* We begin with the following observation. Let $\mathcal{N}$ be a ReLU net, fix a vector $\theta$ of trainable parameters, and consider a neuron $z$ in $\mathcal{N}$. The function $x \mapsto z(x;\theta)$ is continuous and piecewise linear with a finite number of regions $\mathcal{R}$ on which $z(x, \theta)$ is some fixed linear function. On each such $\mathcal{R}$, the gradient $\nabla z$ is constant. If that constant is non-zero, then $H_z(\theta) \cap \mathcal{R}$ is either empty if $b_z$ does not belong to the range of $z$ on $\mathcal{R}$ or is a co-dimension 1 hyperplane if it does. In contrast, if $\nabla z = 0$ on $\mathcal{R}$, then $z$ is constant and $H_z(\theta) \cap \mathcal{R}$ is empty unless $b_z$ is precisely equal to the value of $z$ on $\mathcal{R}$. Thus, given any choice of weights in $\mathcal{N}$ and for all but a finite number of biases, the set $H_z(\theta)$ coincides with co-dimension one hyperplane in each linear region $\mathcal{R}$ for the function $z(\cdot;\theta)$ that it intersects.

Now let us fix all the weights (but not biases) in $\mathcal{N}$ and a collection $z_1, \ldots, z_k$ of $k$ distinct neurons in $\mathcal{N}$ arranged so that $\ell(z_1) \leq \cdots \leq \ell(z_k)$ where $\ell(z)$ denotes the layer of $\mathcal{N}$ to which $z$ belongs. Suppose $x$ belongs to $H_{z_j}(\theta)$ for $j = 1, \ldots, k$ but not to $H_z(\theta)$ for any neuron $z$ not in $\{z_1, \ldots, z_k\}$.

By construction, the function $z_1(\cdot\,;\theta)$ is linear in a neighborhood of $x$. Therefore, by the observation above, near $x$, for all but a finite collection of choices of bias $b_{z_1}$ for $z_1$, $H_{z_1}(\theta)$ coincides with a co-dimension 1 hyperplane. Let us define a new network obtained by restricting $\mathcal{N}$ to this hyperplane (and keeping all the weights from $\mathcal{N}$ fixed). Repeating the preceding argument applied to the neuron $z_2$ in this new network, we find again that, except for a finite number of values for the bias $b_{z_2}$, near $x$ the set $H_{z_2}(\theta) \cap H_{z_1}(\theta)$ is also a co-dimension 1 hyperplane inside $H_{z_1}$. Proceeding in this way shows that for any fixed collection of weights for $\mathcal{N}$, there are only finitely many choices of biases for which the conclusion the present Lemma fails to hold. Thus, in particular, the conclusion holds on a set of probability 1 with respect to any distribution on $\mathbb{R}^{\#\mathrm{params}}$ that has a density relative to Lebesgue measure. $\qquad\square$

Repeating the proof Theorem 6 in [14] with the sets $\widetilde{S}_z$ replaced by $S_z$ (which only makes the proof simpler since Proposition 9 in [14] is not needed) shows that, under the assumptions of Theorem 5, there exists $T > 0$ so that for any bounded $S \subset \mathbb{R}^{n_{\mathrm{in}}}$

$$\frac{\mathbb{E}\left[\mathrm{vol}_{n_{\mathrm{in}}-k}(X_{\mathcal{N},k} \cap S)\right]}{\mathrm{vol}_{n_{\mathrm{in}}}(S)} \;\leq\; T^k \binom{\#\mathrm{neurons}}{k}. \tag{10}$$

We will use the volume bounds in (10) to prove Theorem 5 by an essentially combinatorial argument, which we now explain. Fix a closed cube $\mathcal{C} \subseteq \mathbb{R}^{n_{\mathrm{in}}}$ with sidelength $\delta > 0$ and define

$$\mathcal{C}_k := \text{dimension } k \text{ skeleton of } \mathcal{C}, \qquad k = 0, \ldots, n_{\mathrm{in}}.$$

So for example, $\mathcal{C}_0$ is the $2^{n_{\mathrm{in}}}$ vertices of $\mathcal{C}$, $\mathcal{C}_{n_{\mathrm{in}}}$ is $\mathcal{C}$ itself, and $\mathcal{C}_{n_{\mathrm{in}}-1}$ is the set of $2n_{\mathrm{in}}$ co-dimension 1 faces of $\mathcal{C}$. In general, $\mathcal{C}_k$ consists of $\binom{n_{\mathrm{in}}}{k} 2^{n_{\mathrm{in}}-k}$ linear pieces with dimension $k$, each with volume $\delta^k$. Hence,

$$\mathrm{vol}_k\left(\mathcal{C}_k\right) \;=\; \binom{n_{\mathrm{in}}}{k} 2^{n_{\mathrm{in}}-k} \delta^k \tag{11}$$

For any vector $\theta$ of trainable parameters for $\mathcal{N}$ define

$$\mathcal{V}_k(\theta) \;:=\; X_{\mathcal{N},k}(\theta) \cap \mathcal{C}_k.$$

The collections $\mathcal{V}_k(\theta)$ are useful for the following reason.

**Lemma 12.** *With probability 1 with respect to $\theta$, for every $k$, the set $\mathcal{V}_k(\theta)$ has dimension 0 (i.e. is a collection of discrete points) and*

$$\#\{\mathrm{r}-\text{partial activation regions } \mathcal{R}\left(\mathcal{A};\theta\right) \text{ with } \mathcal{R}\left(\mathcal{A};\theta\right) \cap \mathcal{C} \neq \emptyset\} \;\leq\; \sum_{k=r}^{n_{\mathrm{in}}} \binom{k}{r} 2^{k-r} \#\mathcal{V}_k, \tag{12}$$

*where $\#\mathcal{V}_k$ is the number of points in $\mathcal{V}_k$.*

*Proof.* The statement that generically $\mathcal{V}_k(\theta)$ has dimension 0 follows since, by construction, $\mathcal{C}_k$ has dimension $k$ and, with probability 1, $X_{\mathcal{N},k}(\theta)$ coincides locally with a co-dimension $k$ hyperplane (see Lemma 11) in general position. To prove (12) note that any non-empty, bounded, convex polytope has at least one vertex on its boundary. Thus, with probability 1,

$$\mathcal{R}\left(\mathcal{A};\theta\right) \cap \mathcal{C} \neq \emptyset \quad \Rightarrow \quad \exists k = 0, \ldots, n_{\mathrm{in}} \quad \text{s.t.} \quad \partial\mathcal{R}\left(\mathcal{A};\theta\right) \cap V_k \neq \emptyset,$$

where $\partial\mathcal{R}\left(\mathcal{A};\theta\right)$ is the boundary of the $r$-partial activation region $\mathcal{R}\left(\mathcal{A};\theta\right)$. Indeed, if an $r$-partial activation region $\mathcal{R}\left(\mathcal{A};\theta\right)$ intersects $\mathcal{C}$, then $\mathcal{R}\left(\mathcal{A};\theta\right) \cap \mathcal{C}$ is a non-empty, bounded, convex polytope. It must therefore have a vertex on its boundary. This vertex is either in the interior of $\mathcal{C}$ (and hence belongs to $\mathcal{V}_{n_{\mathrm{in}}}$) or is the intersection of some co-dimension $k$ part of the boundary of $\mathcal{R}\left(\mathcal{A};\theta\right)$, which belongs with probability 1 to $X_{\mathcal{N},r+k}$, with some $r+k$-dimensional face of the boundary of $\mathcal{C}$ (and hence belongs to $\mathcal{V}_{k+r}$). Thus, with probability 1,

$$\#\{\mathrm{r}\text{-partial activation regions } \mathcal{R}\left(\mathcal{A};\theta\right) \text{ with } \mathcal{R}\left(\mathcal{A};\theta\right) \cap \mathcal{C} \neq \emptyset\} \;\leq\; \sum_{k=r}^{n_{\mathrm{in}}} T_k \#\mathcal{V}_k,$$

where $T_k$ is the maximum over all $v \in \mathcal{V}_k$ of the number of $r$-partial activation regions whose boundary contains $v$. To complete the proof, it remains to check that, with probability 1 over $\theta$,

$$\forall v \in \mathcal{V}_k \qquad \#\{\mathrm{r}-\text{partial regions } \mathcal{R}\left(\mathcal{A};\theta\right) \mid v \in \partial\mathcal{R}\left(\mathcal{A};\theta\right)\} \;\leq\; \binom{k}{r} 2^{k-r}.$$

To see this, note that, by definition of $X_{\mathcal{N},k}$, in a sufficiently small neighborhood $U$ of any $v \in \mathcal{V}_k$, all but $k$ neurons have pre-activations satisfying either $z(x) > b_z$ or $z(x) < b_z$ for all $x \in U$. Thus, there are at most $\binom{k}{r} 2^{k-r}$ different $r$-partial activation patterns $\mathcal{A}$ whose $r$-partial activation region $\mathcal{R}(\mathcal{A}; \theta)$ can intersect $U$. $\qquad\square$

**Lemma 13.** *Fix $k = 0, \ldots, n_{\mathrm{in}}$. For all but a measure $0$ set of vectors $\theta$ of trainable parameters for $\mathcal{N}$ there exists $\varepsilon > 0$ (depending on $\theta$) so that the balls $B_\varepsilon(v)$ of radius $\varepsilon$ centered at $v \in \mathcal{V}_k$ are disjoint and*

$$\mathrm{vol}_{n_{\mathrm{in}}-k}\left(X_{\mathcal{N},k} \cap B_\varepsilon(v)\right) = \varepsilon^{n_{\mathrm{in}}-k}\omega_{n_{\mathrm{in}}-k},$$

*where $\omega_k$ is the volume of unit ball in $\mathbb{R}^k$.*

*Proof.* With probability $1$ over $\theta$, each $\mathcal{V}_k$, by Lemma 12, is a discrete collection of points. Hence, we may choose $\varepsilon > 0$ sufficiently small so that the balls $B_\varepsilon(v)$ are disjoint. Moreover, by Lemma 11, in a sufficiently small neighborhood of $v \in \mathcal{V}_k$, the set $X_{\mathcal{N},k}$ coincides with a $(n_{\mathrm{in}} - k)$-dimensional hyperplane passing through $v$. The $(n_{\mathrm{in}} - k)-$dimensional volume of this hyperplane in $B_\varepsilon(v)$ is the volume of an $(n_{\mathrm{in}} - k)-$dimensional ball of radius $\varepsilon$, which equals $\varepsilon^{n_{\mathrm{in}}-k}\omega_{n_{\mathrm{in}}-k}$, completing the proof. $\qquad\square$

For each $k = 0, \ldots, n_{\mathrm{in}}$, denote by

$$\mathcal{C}_{k,\varepsilon} := \left\{ x \in \mathbb{R}^{n_{\mathrm{in}}} \mid \mathrm{dist}(x, \mathcal{C}_k) \leq \varepsilon \right\}$$

the $\varepsilon$- thickening of $\mathcal{C}_k$. Then Lemma 13 implies that for all but a measure $0$ set of $\theta \in \mathbb{R}^{\#\mathrm{params}}$, there exists $\varepsilon > 0$ so that

$$\frac{\mathrm{vol}_{n_{\mathrm{in}}-k}\left(X_{\mathcal{N},k} \cap \mathcal{C}_{k,\varepsilon}\right)}{\varepsilon^{n_{\mathrm{in}}-k}\omega_{n_{\mathrm{in}}-k}} \geq \#\mathcal{V}_k.$$

Indeed, for any $\theta$ in a full measure set, Lemma 13 guarantees the existence of an $\varepsilon > 0$ so that the $\varepsilon$ balls $B_\varepsilon(v)$ are contained in $\mathcal{C}_{k,\varepsilon}$ and are disjoint. Hence,

$$\mathrm{vol}_{n_{\mathrm{in}}-k}\left(X_{\mathcal{N},k} \cap \mathcal{C}_{k,\varepsilon}\right) \geq \sum_{v \in \mathcal{V}_k} \varepsilon^{n_{\mathrm{in}}-k}\omega_{n_{\mathrm{in}}-k} = \#\mathcal{V}_k \cdot \varepsilon^{n_{\mathrm{in}}-k}\omega_{n_{\mathrm{in}}-k}.$$

Note that

$$\lim_{\varepsilon \to 0} \frac{\mathrm{vol}_{n_{\mathrm{in}}}\left(\mathcal{C}_{k,\varepsilon}\right)}{\varepsilon^{n_{\mathrm{in}}-k}\omega_{n_{\mathrm{in}}-k}} = \mathrm{vol}_k\left(\mathcal{C}_k\right).$$

Thus, taking $\varepsilon \to 0$, we have

$$\mathbb{E}\left[\#\mathcal{V}_k\right] \leq \mathbb{E}\left[\lim_{\varepsilon \to 0} \frac{\mathrm{vol}_k\left(X_{\mathcal{N},k} \cap \mathcal{C}_{k,\varepsilon}\right)}{\varepsilon^{n_{\mathrm{in}}-k}\omega_{n_{\mathrm{in}}-k}}\right].$$

By Fatou's Lemma and (10) there exists $T > 0$ such that

$$\mathbb{E}\left[\#\mathcal{V}_k\right] \leq \lim_{\varepsilon \to 0} \mathbb{E}\left[\frac{\mathrm{vol}_{n_{\mathrm{in}}-k}\left(X_{\mathcal{N},k} \cap \mathcal{C}_{k,\varepsilon}\right)}{\varepsilon^{n_{\mathrm{in}}-k}\omega_{n_{\mathrm{in}}-k}}\right] \leq T^k \binom{\#\mathrm{neurons}}{k} \mathrm{vol}_k(\mathcal{C}_k).$$

Combining this with (11) and (12), we find

$$\mathbb{E}\left[\#\{\mathrm{r-partial\ activation\ regions}\ \mathcal{R}\left(\mathcal{A}; \theta\right)\ \mathrm{with}\ \mathcal{R}\left(\mathcal{A}; \theta\right) \cap \mathcal{C} \neq \emptyset\}\right]$$

is bounded above by

$$\sum_{k=r}^{n_{\mathrm{in}}} \binom{n_{\mathrm{in}}}{k}\binom{\#\mathrm{neurons}}{k}\binom{k}{r} 2^{k-r} 2^{n_{\mathrm{in}}-k} (\delta T)^k \leq 2^{2n_{\mathrm{in}}-r} \sum_{k=r}^{n_{\mathrm{in}}} \binom{n_{\mathrm{in}}}{k}\binom{\#\mathrm{neurons}}{k}(\delta T)^k$$

$$\leq 2^{-r}(4T\delta)^{n_{\mathrm{in}}} \sum_{k=r}^{n_{\mathrm{in}}} \binom{n_{\mathrm{in}}}{k}^2 \frac{\binom{\#\mathrm{neurons}}{k}}{\binom{n_{\mathrm{in}}}{k}},$$

where in the last line we've assumed $\delta > 1/T$ and in the first inequality we used that

$$\binom{k}{r} \leq \sum_{r=0}^{k} \binom{k}{r} = 2^k \leq 2^{n_{\text{in}}}.$$

Observe that

$$\frac{\binom{\#\text{neurons}}{k}}{\binom{n_{\text{in}}}{k}} = \frac{(\#\text{neurons})! \cdot (n-k)!}{n_{\text{in}}! \cdot (\#\text{neurons}-k)!} \leq \frac{(\#\text{neurons})^{n_{\text{in}}}}{n_{\text{in}}!} \cdot \frac{(n_{\text{in}}-k)!}{(\#\text{neurons})^{n_{\text{in}}-k}} \leq \frac{(\#\text{neurons})^{n_{\text{in}}}}{n_{\text{in}}!}.$$

Hence, using that

$$\sum_{k=0}^{n_{\text{in}}} \binom{n_{\text{in}}}{k}^2 = \binom{2n_{\text{in}}}{n_{\text{in}}} \leq 4^{n_{\text{in}}},$$

$\mathbb{E}\left[\#\{\text{activation regions } \mathcal{R}(\mathcal{A};\theta) \text{ with } \mathcal{R}(\mathcal{A};\theta) \cap \mathcal{C} \neq \emptyset\}\right]$ is bounded above by

$$\frac{(T\#\text{neurons})^{n_{\text{in}}}}{2^r n_{\text{in}}!} \text{ vol} (\mathcal{C}).$$

$\square$

## D  Zero Bias

The reader may notice that, as stated, Theorem 5 does not apply for networks with zero biases, as is commonly the case at initialization. While zero biases clearly do not persist during normal training past initialization, it is interesting to consider how many activation regions occur in this case. No finite value of constant $C_{\text{bias}}$ in Theorem 5 satisfies Condition 4; does this mean that the number of activation regions goes to infinity? The answer is no:

**Proposition 14.** *Suppose that $\mathcal{N}$ is a deep ReLU net with any bias distribution, and let $\mathcal{N}_0^{bias}$ be the same network with all biases set to $0$. Then, the total number of activation regions (over all of input space) for $\mathcal{N}_0^{bias}$ is no more than that for $\mathcal{N}$.*

A key point in the proof is the scale equivariance of ReLU networks with zero bias.

**Lemma 15.** *Let $\mathcal{N}$ be a ReLU network with biases set identically to zero. Then,*

  (a) *$\mathcal{N}$ is equivariant under positive constant multiplication: $\mathcal{N}(cx) = c\mathcal{N}(x)$ for each $c > 0$.*

  (b) *For every activation region $R$ of $\mathcal{N}$, and every point $x$ in $R$, all points $cx$ are also in $R$ for $c > 0$ (this implies that $R$ is a convex cone).*

*Proof.* Note that each neuron of the network computes a function of the form $z(x_1, \ldots, x_m) = \text{ReLU}(\sum_{i=1}^{m} w_i x_i)$. Note that:

$$z(cx_1, \ldots, cx_m) = \text{ReLU}\left(c \sum_{i=1}^{m} w_i x_i\right) = c \cdot \text{ReLU}\left(\sum_{i=1}^{m} w_i x_i\right) = cz(x_1, \ldots, x_m).$$

Thus, each neuron is equivariant under multiplication by positive constants $c$, and thus the overall network must be as well, proving (a). Note also that the activation patterns of ReLUs for $x$ and $cx$ must also be identical, implying that $x$ and $cx$ lie in the same linear region. This proves (b).

Wee now turn to the proof of Proposition 14. We proceed by defining an injective mapping from regions of $\mathcal{N}_0^{bias}$ to regions of $\mathcal{N}$. For each linear region $R$ of $\mathcal{N}_0^{bias}$, pick a point $x_R \in R$. By the Lemma, $cx_R \in R$ for each $c > 0$. Let $\mathcal{N}_{1/c}^{bias}$ be the network obtained from $\mathcal{N}$ by dividing all biases by $c$, and observe that $\mathcal{N}(cx) = c\mathcal{N}_{1/c}^{bias}(x)$, with the same activation pattern between the two networks. But by picking $c$ sufficiently large, $\mathcal{N}_{1/c}^{bias}$ becomes arbitrarily close to $\mathcal{N}_0^{bias}$. We conclude that for some sufficiently large $c_R$, $\mathcal{N}_0^{bias}(c_R x_R)$ and $\mathcal{N}(c_R x_R)$ have the same pattern of activations, so the regions of $\mathcal{N}$ in which $c_R x_R$ lies must be distinct for all distinct $R$. Thus, the number of regions of $\mathcal{N}$ must be at least as large as the number of regions of $\mathcal{N}_0^{bias}$, as desired. $\square$

An informal argument suggests that Proposition 14 is relatively weak. As a consequence of Lemma 15(b), for any zero-bias net $\mathcal{N}_0^{\text{bias}}$ and any closed surface $S$ containing the origin, $S$ must intersect every linear region of $\mathcal{N}_0^{\text{bias}}$ (since such regions are convex cones with apices at the origin). Taking $S$ to be the unit hypercube in $n_{\text{in}}$ dimensions, the total number of activation regions of $\mathcal{N}_0^{\text{bias}}$ is no more than the sum of the numbers of activation regions intersecting each of its $2n_{\text{in}}$ facets, each of dimension $(n_{\text{in}} - 1)$. We expect from Theorem 5 that the number of activation regions intersecting each facet is like $(\#\text{neurons})^{n_{\text{in}}-1}$. Thus, the total number of activation regions of $\mathcal{N}_0^{\text{bias}}$ should grow no faster than $n_{\text{in}} (\#\text{neurons})^{n_{\text{in}}-1}$, instead of the $n_{\text{in}} (\#\text{neurons})^{n_{\text{in}}}$ directly implied by Proposition 14.

Lemma 15 implies that networks with small bias are *almost* scale equivariant. Figure 7 shows the activation regions for a network initialized with very small biases (i.i.d. normal with variance $10^{-6}$). Part (a) displays regions intersecting a plane through the origin, indicating that, outside of a small radius around the origin, all regions are infinite and are approximated by cones. Thus, ReLU networks near initialization are almost scale equivariant except for sample points very close to the origin, a simple but potentially useful property that has not been widely recognized. During training, biases grow and the radius grows within which finite regions occur. Note that, as shown in part (b) of the figure, a plane not containing the origin does not reveal this structure, as such a plane can have finite intersection with many regions that are in fact infinite.