[Reviews · NeurIPS 2019]

Reviewer 1



In general I like the paper. The definition of activation regions is sound and it makes intuitive sense. The empirical evaluations are also well thought out. Specific comments: The authors state that usually there is no distinction between activation and linear regions. The way activation regions are defined in the paper, there is no reason to believe the two to be identical. Along the same lines, the fact that one is a union of the other (Lemma 3) is expected. While I understand and buy into the definition of an activation region, I think more intuition would be helpful. For example, why it suffices that A is arbitrary, why -1 to the power of a_z, etc. In (3), activation regions is not defined. I believe it is the union of individual activation regions. Line 39: it's unclear what is T at this point. It should be added: for a constant T. Theorem 5: conditions 1 and 2 should be simply restated as continuous RVs. Line 177: two heuristics for the second (5) - there is just one (5) In 3.2, the intuition is a little bit too shaky. Between "many times" and O(1) there are many options in between. In other words, if something is not 'many times' I would not conclude that it is O(1). F_learn is vaguely introduced, but no further discussion is provided. I find this confusing and suggest to remove the references to this set. There was work in the last ICML (Long Beach) on linear regions. If it is not cited, it should be added and discussed.

Reviewer 2



General comment: It is a nice addition to the previous paper [14] in that it actually counts the number of activation regions other than just the n-k volumes of the discontinuity points of the gradient (which these results use in the proof). Compared to previous works of counting linear regions, this paper looks at the number when randomly drawing the weights and biases (not necessarily independent). It is well written However it could benefit from some more intuitive explanations at times. For example, I have some comments that would be nice to see addressed in a revised version of the paper: - It is not very clearly stated in the paper (terms like "strongly suggests" are used without being alluded to after the rigorous theorem statement) how much training can you actually do so that the bound still holds and under which circumstances would it break down? By definition of a density C_bias seems to be <= 1, so that it wouldn't matter in the grand scheme of things but potentially get a tighter bound? Perhaps the authors are already working on this, but I'm curious: What are the challenges to actually show how the number of regions evolves over time with gradient descent? - Related to this point: In the experiments, the actual number of regions is higher than predicted but you don't offer an explanation. d/n < 1, thus C_grad is probably < 1 given the result in [12], and as C_bias should be <= 1, what does make the number grow so much during training? I find it nice however that at initialization, the number of regions pretty much matches the theoretical bound (could draw a horizontal line for theoretical prediction in the plot perhaps). - I am not sure why Lemma 6 i.e. invariance of the count to uniform bias scaling would actually imply invariance to non-uniform bias scaling which is in turn equivalent to uniform scaling of weights? You do mention that you don't prove it, but why should that be true intuitively? In general, scaling of the randomized weights seems to have rather big impacts of generalization properties of the final neural networks which is what makes me feel uncertain about this claim. - I think it is important to reveal the dependence on d/n through C_grad, C_bias - i.e. that they may grow with d/n / correlation but are "constant" in terms of number of neurons. You later state in 3.3 that "depth does not increase the local density of activation regions" - this sentence is misleading as this is perhaps approximately true for wide networks but strictly speaking not in general Minor comments for improvement: - It would be useful to explicitly state the results of previous papers on the max number of linear regions so that the reader can see the difference directly. - A figure illustrating bent hyperplanes would be very useful - Maybe it would be nice to define "fan-in" somewhere? - in 3.2. you write I and sometimes [a,b]

Reviewer 3



Definitions 1-2 and Lemmas 1-4 seem clear. The paper studies upper bounds on the number of linear regions of the functions represented by ReLU neural networks over a box of the input space, given certain conditions on the gradients of the neurons and the biases. The idea is to draw a distinction between the theoretically possible number and the number that might be observed in practice. The main result discusses how under certain conditions on parameters and gradients, the number of linear regions of the function represented by the network is usually small. Although the paper interprets the result as `depth independent', it is presented in a way that includes constants that may well be influenced by depth. The theorem could be interpreted as saying that there are regions of parameter space where the number of regions of the represented functions is small. This is not surprising, as claimed in the title of the paper. Nonetheless, the result is interesting in that it gives some details on the regions of the parameter space where this happens, even if this description includes somewhat less explicit conditions on gradients of neurons. The paper also seeks to explain how the requirements of the theorem might hold true at initialization and during training. It presents some experiments evaluating the linear regions observed along one dimensional paths on input space.

[Author Response · NeurIPS 2019]

**Response to Reviewer** #1: Thank you for the careful reading and feedback. In a revision, we will address the detailed comments:

1. We will clarify the discussion around Lemma 3 to reflect that activation regions and linear regions are typically identified in prior literature but are in fact not quite the same.
2. We will add intuitive explanations of the +/- 1s and activation patterns in the definition of activation regions.
3. Around (3), we will clarify that $\text{activationregions}(\mathcal{N}, \theta)$ is the set of all non-empty activation regions for the network $\mathcal{N}$ with trainable parameters $\theta$.
4. We will emphasize in line 39 that $T$ is constant.
5. We will restate conditions 1,2 in terms of continuous random variables.
6. We will correct the reference to (5) in line 177.
7. We will sharpen the intuition in Section 3.2 to reflect that $|z'(x)| = O(1)$ guarantees $O(1)$ high amplitude oscillations for $z(x)$ when $x$ varies over a fixed bounded interval.
8. We will make it clearer that this terminology is deliberately vague - the terms $F_{learn}$ etc are only referenced in the caption to Figure 1.
9. We are not sure what work from ICML 2019 the reviewer has in mind. In case Hanin and Rolnick was meant here, we do make a point of citing it as [14].

**Response to Reviewer** #2: Thank you for the careful reading and feedback. About point 3 in the reviewer's list of three contributions: we found the fact that networks cannot learn many activation regions to be a surprising counterpoint to the well-known ability of networks to memorize high-dimensional noise. We plan to amplify this point in the revision. In the revision, we will also address the reviewer's detailed comments:

1. We agree that more intuition can be helpful and plan to add more (see points 1,2,7 in our response to Reviewer #1).
2. We will give a more thorough discussion of the constants $C_{grad}$, $C_{bias}$. Previous work [11,12] shows that $C_{grad}$ is like $d/n$ only at init, and hence can in principle grow through training, as the reviewer suggests. It is not clear how to rule this out *a priori*.
3. About Lemma 6, we agree that our discussion could be clarified and will write simply that "we conjecture" that the inhomogeneous scaling of biases does not strongly affect the number of regions.
4. We agree that a discussion of which architectures have large $C_{grad}$ is warranted. We will explain that prior work [11,12,13] shows that unless $C_{grad}$ is small, fully connected ReLU nets have unstable forward and backward passes at init. Thus, for such networks, as long as they are trainable, $C_{grad}$ will not be too large. This is the reason we used terms like "depth-independent", and we will amplify this point.

**Response to Reviewer** #3: Thank you for the careful reading and feedback. About the reviewer's comment that some of our experiments could be seen as illustrations rather than empirical evidence: we will emphasize in the revision that, indeed, at init, they are simply illustrations of our results. However, after init, it is not clear how $C_{grad}$, $C_{bias}$ behave and hence empirical validation that our results apply is provided by these experiments. In the revision, we will also address the reviewer's detailed comments:

1A. We agree that Definitions 1-2 and Lemmas 1-4 are elementary, and their purpose is primarily for clarity in exposition. Moreover, we wanted to give a clear delineation between linear and activation regions, which have often been conflated in prior work.
2A. We agree that the potential dependence of $C_{grad}$ on depth needs to be discussed. See points $2, 4$ in our response to Reviewer #2.

About the reviewer's suggestions on how to improve:

1B. Our results show both theoretically and empirically that not only *can* the number of regions be small but that it typically *is* small both at init and throughout training. We believe this is an important point and will emphasize it in the revision.
2B. In the revision we will emphasize that although our results do not directly influence architecture selection, they make more clear the role of depth and hence suggest to practitioners the intuition that network depth is mainly useful for optimization and not for expressivity.
3B. See point 2A above and point 4 in our response to Reviewer #2.

[Meta-Review · NeurIPS 2019]

This paper provides evidence that the number of possible activations is quite small, in stylized settings (it seems the paper is based mainly on remark 1). The paper is interesting and appears to be the first most concrete evidence of this, though to some extent this is implied by ntk analyses; I ask that the authors provide a concrete technical comparison to ntk and related papers/consequences. I also ask for a more thorough comparison to VC dimension results, which also proceed via region (and activation) counting.